# Single-Cell Analysis with Silver-Coated Pipette by Combined SERS and SICM

**DOI:** 10.3390/cells12212521

**Published:** 2023-10-25

**Authors:** Sergey Dubkov, Aleksei Overchenko, Denis Novikov, Vasilii Kolmogorov, Lidiya Volkova, Petr Gorelkin, Alexander Erofeev, Yuri Parkhomenko

**Affiliations:** 1Institute of Advanced Materials and Technologies, National Research University of Electronic Technology, 124498 Moscow, Russia; 2Research Laboratory of Biophysics, National University of Science and Technology “MISIS” (MISIS), 119049 Moscow, Russiapg@icappic.com (P.G.); erofeev@icappic.com (A.E.);; 3Molecular Nanophotonics Group, Peter Debye Institute for Soft Matter Physics, Leipzig University, 04109 Leipzig, Germany; 4Department of Chemistry, Lomonosov Moscow State University, 119991 Moscow, Russia; 5Institute of Nanotechnology of Microelectronics RAS, 115487 Moscow, Russia

**Keywords:** pipette, SICM, SERS, Raman spectroscopy, Young’s modulus, Ag nanoparticles

## Abstract

The study of individual cell processes that occur both on their surface and inside is highly interesting for the development of new medical drugs, cytology and cell technologies. This work presents an original technique for fabricating the silver-coated pipette and its use for the cell analysis by combination with surface-enhanced Raman spectroscopy (SERS) and scanning ion-conducting microscopy (SICM). Unlike the majority of other designs, the pipette opening in our case remains uncovered, which is important for SICM. SERS-active Ag nanoparticles on the pipette surface are formed by vacuum–thermal evaporation followed by annealing. An array of nanoparticles had a diameter on the order of 36 nm and spacing of 12 nm. A two-particle model based on Laplace equations is used to calculate a theoretical enhancement factor (EF). The surface morphology of the samples is investigated by scanning electron microscopy while SICM is used to reveal the surface topography, to evaluate Young’s modulus of living cells and to control an injection of the SERS-active pipettes into them. A Raman microscope–spectrometer was used to collect characteristic SERS spectra of cells and cell components. Local Raman spectra were obtained from the cytoplasm and nucleus of the same HEK-293 cancer cell. The EF of the SERS-active pipette was 7 × 10^5^. As a result, we demonstrate utilizing the silver-coated pipette for both the SICM study and the molecular composition analysis of cytoplasm and the nucleus of living cells by SERS. The probe localization in cells is successfully achieved.

## 1. Introduction

In recent decades, probe microscopy techniques have made a great progress in scanning living cells with the ability to visualize individual molecules in vivo at nanometer resolution [1]. However, these methods have certain drawbacks. Fluorescence imaging requires the use of markers that can negatively affect the accuracy of cellular machinery detection, and the atomic force microscopy (AFM) probe has some limitations for long-term measurements [2]. The combination of a fluorescence-based imaging platform and SERS is frequently employed for researching cell functional characteristics and their interactions with nanomaterials [3,4,5]. However, the resolution of such methods is limited by half of the excitation wavelength. The SERS application of SICM based on a silver-coated nanopipette provides the possibility of scanning topography with a nanoscale resolution prior to the SERS sampling procedure. The feedback control of SICM based on ion current registration [6] allows for the performance of SERS sampling on different compartments of living cells with a highly controlled depth. Also, in our previous work [7], we demonstrated SICM application for Young’s modulus estimation of living cells based on the intrinsic force between the nanopipette tip and cell surface. In this work, we revealed that the same characterization of local mechanical properties can be performed using a silver-coated nanopipette for SERS.

Unlike AFM, scanning ion conducting microscopy (SICM) has the advantage of non-contact topographic scanning [8]. SICM has established itself as a tool for scanning live cancer cells, neurons and proteins with a nanoscale resolution via the hopping mod [9,10]. In [11], a comparison of the SICM and AFM methods was made in visualizing such a complex object for probe microscopy as collagen fibrils. The studies of this group demonstrate the high potential of SICM for the non-contact scanning of developed structures in opposition to AFM.

As demonstrated earlier, SICM provides a local estimation of Young’s modulus of living cells with less force applied to the cell surface and, consequently, less deformation of its surface. Young’s modulus of living cells determines such biological functions as division, growth, differentiation, mobility and tissue homeostasis [12]. SICM provides long scanning of cell topography and estimation of Young’s modulus distribution. Furthermore, using correlative confocal imaging of the cytoskeleton, we showed that the estimated Young’s modulus is directly related to the state of the cytoskeleton. The developed technique was then successfully used to characterize the cytoskeleton state under different conditions [13,14,15].

In addition to Young’s modulus, the internal compound and products of vital activity of living cells are determining characteristics [16]. The ability to study these parameters, to obtain mechanical characteristics and topographic maps of living cells at the same time, will significantly expand the SICM toolkit. It is possible to obtain data about molecular composition of the cell parts using surface-enhanced Raman scattering (SERS) spectroscopy, whose unique sensitivity is known to provide detection and identification of single molecules [17,18]. In this case, a pipette plays the role of an SERS-active tool [19,20,21]. SERS techniques have also been actively used to study single cells of plants and bacteria [22,23]. To impart SERS activity to the pipette, its tip is modified with plasmonic metals such as Au, Ag and Cu [24]. It is important to note that the surface of the pipette is a shrink cone with a high aspect ratio, which makes the process of forming an array of nanoparticles on such a surface poorly controlled [25]. One of the most widely used methods of nanoparticle array formation at the tip of the pipette is the chemical method [26,27,28,29], the main disadvantage of which is full or partial overlapping of the pipette opening [26,27,28]. The principle of SICM is based on the registration of ionic current through the opening at the pipette tip [6]. The ionic current decreases as the pipette approaches the surface at the level of the pore distance, allowing the distance between the pipette and the surface to be adjusted to scan the surface in the electrolyte solution in a non-contact mode. The non-contact mode and minimal surface voltage are very important for studying living cells such as neurons. In case of partial overlap of the pipette opening, the functioning of SICM and methods based on it are unstable, and in case of complete overlap, they are impossible.

However, previous works have proposed the chemical method [19,21,29,30]. The chemical method also has other disadvantages such as low size reproducibility, long duration of deposition and difficulty in controlling the particle size [26,27,28,29]. An immersion deposition, which is a derivative of the chemical approach, has been exploited for the careful adjustment of the geometrical parameters of metal nanoparticles [31,32]. However, it cannot be considered as an appropriate technique if a glass-based substrate (e.g., glass pipette) is used. An alternative method for forming arrays of nanoparticles on a surface is a physical deposition, such as vacuum–thermal evaporation followed by annealing. This method has high reproducibility, controllability of the process to control the size of nanoparticles, and is one of the simplest and most effective methods for forming metallic nanoparticles and films [33,34].

The pipette modified with plasmonic metals can be used to regulate the delivery of molecules/ions and to study in vivo their effects on a living cell by Raman spectroscopy. Such a pipette would enable the study of the elastomeric characteristics of cells. The present SICM tooling will allow point injection of the modified pipette inside the cell or to the cell surface. Due to the nanoscale tip of the pipette (10–100 nm), it has a minimal effect on the cell under study when it is inserted into the cell [35]. A similar cone-shaped structure which acts as a rod-like antenna with a special surface plasmon resonance property is mentioned in work [36].

In this study, we have developed a technique for the formation of an array of Ag nanoparticles by vacuum–thermal evaporation onto a rotating pipette for SICM studies of living cells and for the accurate differentiation of the pipette location in cell and ultrasensitive detection of anomalies in molecular composition of membrane, cytoplasm and nucleus depending on a target cell line or external effects like interactions with drugs and temperature variation, to name a couple. In previous works [13,14] we showed that SICM can be successfully used for characterization of novel anticancer drugs such as cytostatics. Because the mechanism of cytostatics involves a change in the cytoskeleton state, the efficiency of a novel medicine can be estimated by measuring the cell’s Young’s modulus. Thus, SERS specific analytes can be characterized simultaneously in a single cell using a combination of silver-coated nanopipettes.

## 2. Materials and Methods

### 2.1. Fabrication of Pipettes

Borosilicate glass tubes (OD: 1.0 mm; ID: 0.5 mm) (WPI, Hitchin, Hertfordshire, UK) were used for the pipette fabrication. The following two-step program was used to make pipettes with a typical tip radius of 60 nm using the laser puller P-2000 (Sutter Instruments, Novato, CA, USA) [7]:

Heat 310, Fil 3, Vel 30, Del 160, Pul 0.

Heat 330, Fil 3, Vel 25, Del 160, and Pul 200.

The tip radius of the fabricated pipettes was estimated using the following theoretical model [37,38]:(1)$$r=\frac{Io}{\pi kVtan(\alpha )},$$
where the half-cone angle *α* is 3 degrees, *k* is 1.35 S m^−1^ and *V* is the holding potential of 200 mV.

### 2.2. Formation of Ag Nanoparticle Array

Metal nanoparticles were formed by vacuum–thermal evaporation (MVU TM TIS 02, NIITM, Moscow, Russia) followed by heat treatment. Ag target was used as a source for the plasmonic material evaporation. The chamber residual pressure was approx. 3 × 10^−5^ Torr. The annealing of samples was carried out in vacuum at a pressure of 3 × 10^−5^ Torr at 230 °C for 30 min. Ag evaporation was performed on a rotating pipette. A 25 × 38 mm collector electric motor was used as the moving element. A schematic representation of the nanoparticle formation technique at the tip of the pipette is shown in Figure 1a. Before and after the nanoparticle deposition process, the pipettes were stored in vacuum at room temperature to prevent the contamination of the pipette, or degradation of the particle array [39]. The vacuum box with modified nanopipettes was opened immediately before each measurement with cells. In this way, we avoided surface composition changes caused by air and provided the same storage conditions prior the measurements.

### 2.3. Characterization of the Surface Morphology of the Samples

Helios C4GX scanning electron microscope (Thermo Fisher Scientific, Waltham, MA, USA) was used to study the morphology of the obtained arrays of silver nanoparticles. Imaging of the modified pipette samples was performed at an accelerating voltage of 1 kV and a current of 16 pA at a magnification of 160,000×. The obtained SEM images were analyzed using ImageJ 1.49v. Statistical analysis of the geometrical parameters of Ag nanoparticle arrays was performed using standard mathematical tools—mean nanoparticle diameter (d), mean nanoparticle spacing (l), standard deviation (σ), relative standard deviation for d (σ/d) and for l (σ/l).

### 2.4. Cell Culture

HEK-293 cells (ATCC, Manassas, VT, USA) were cultured in DMEM F12 medium (Gibco, Billings, Montana, MO, USA). Then, cells were reattached from culture flask by using phosphate-buffered solution (PBS) without Ca^2+^ and Mg^2+^ and Trypl-E. Then, cells were cultured in a 35 mm Petri dish (Corning, Corning, NY, USA) for 24 h in DMEM F-12 medium. For nucleus staining, Hoechst 33342 (Thermo Fisher, Waltham, MA, USA) was used at a final concentration of 1 g/mL for 20 min at 37 °C and 5% CO_2_. Before scanning, cells were washed three times with Hank’s solution (HBSS, Lowell, MA, USA). For SERS, cells were cultured on cover glass. After incubation for 24 h, cover glass with HEK-293 cells was placed on the SERS-active substrate to collect the spectrum.

### 2.5. Scanning Ion-Conductance Microscopy (SICM)

SICM made by ICAPPIC (ICAPPIC Limited, London, UK) was used for non-contact topographic scanning and estimation of Young’s modulus of living HEK-293 cells with non-modified and SERS-active pipettes. The scanning platform was mounted on inverted optical microscope Nikon Ti-2 (Nikon, Tokyo, Japan). For estimating Young’s modulus of living cells, the pipette was moved to the surface until the ion current through the tip reduced by 2% from its initial value during scanning. A non-contact topographic image was obtained at an ion current decrease of 0.5% and a further two images were obtained at an ion current decrease (or set-point) of 1% and 2% corresponding to membrane deformations produced by intrinsic force at each setpoint. Young’s modulus was calculated using the previously described theoretical model [40,41]:(2)$${E=PA(\frac{Ssub}{Scell}-1)}^{-1},$$
where *E* is the estimated Young’s modulus, *P* is the applied pressure, *A* is a constant depending on the pipette geometry, and *S_sub_* and *S_cell_* are the slopes of the current–distance curve observed between the ion current decreases of 1% and 2% at the non-deformable surface (*S_sub_*—substrate) and cell surface (*S_cell_*), respectively.

For local SERS measurements, the topography of HEK-293 cells was scanned in non-contact mode. To distinguish the nucleus from the cytoplasm, the fluorescent dye Hoechst was used. Also, SERS spectra were obtained from Hoechst to identify it from the living cell nucleus sampled by the SERS pipette.

The cells were penetrated at a 1 µm depth after the pipette had been positioned on the nucleus area and cytoplasm. After 10 min, the pipette was removed from the cell and the SERS spectrum was measured at the pipette tip.

### 2.6. Raman and SERS Measurements

Raman and SERS spectra were obtained using 3D scanning laser confocal microscope–spectrometer Confotec MR200 (SOL Instruments, Minsk, Belarus) that provides a 3 cm^−1^ spectral resolution. As a test analyte, aqueous solutions of rhodamine 6 G (R6G) were used [42] with concentrations from 1 × 10^−1^ M to 5 × 10^−8^ М. R6G molecules were adsorbed on the pipette surface by its immersion into the analyte solution for 10 min followed by air drying for 10 min. To detect R6G at different concentrations, a new pipette was used each time. A laser with a wavelength of 532 nm and a 40× objective were used in the study. The laser was focused on the pipette tip perpendicular to the axis of its symmetry. All the spectra were captured at an accumulation time of 1 s and a power of ~0.03 mW.

A planar SERS-active substrate was used to obtain SERS spectra of Hoechst 33342 dye and HEK-293 [33], representing a multilayer structure: a substrate of single-crystal oxidized silicon, a silver layer ~100 nm thickness, a SiO_2_ layer ~20 nm thickness, silver nanoparticles with a diameter of 40 nm and a distance between the particles of about 35 nm. Hoechst 33342 was applied to the substrate using a micropipette (ECOCHEM, St. Petersburg, Russia) at a concentration of about 10 µg/mL. HEK-293 cells cultured on a glass coverslip were brought into contact with the SERS-active substrate by pressing. Cell adhesion was provided by gravity. SERS spectra were measured 10 min after preparation of the SERS-active substrate sandwich/coverslip [43].

### 2.7. Calculation of the Enhancement Factor (EF)

To calculate the theoretical enhancement factor (EF) of the electromagnetic component of SERS, the equation from [44] based on Laplace transformations was used:(3)$$EF={\left[\varepsilon \left(\omega \right)/(\varepsilon_{1}+\varepsilon \left(\omega \right))\right]}^{4}{\left(\frac{34.87}{\eta_{0}^{5}}\right)}^{2}={\left[\varepsilon \left(\omega \right)/(\varepsilon_{1}+\varepsilon \left(\omega \right))\right]}^{4}\left(\frac{1216}{\eta_{0}^{10}}\right),$$
where $\varepsilon \left(\omega \right)$—dielectric permittivity of a metallic particle under radiation with a certain wavelength; $\varepsilon_{1}$—average dielectric constant of the surrounding environment; $\eta_{0}$—critical parameter for spherical dimers, which is calculated as shown below:(4)$$\eta_{0}=\sqrt{\frac{2\delta }{R}}=\sqrt{\frac{d}{R}},$$
where δ—distance from the particle surface to the hot spot center; d=2δ—the shortest distance between two spherical particles.

To calculate $\varepsilon \left(\omega \right)$, the values from [45] for 532 nm were applied. For $\varepsilon_{1}$, the average value of dielectric permittivity of borosilicate glass [46] and air was used.

Calculation of the enhancement factor for the modified pipettes obtained during the experiments was performed using the formula [47]:(5)$$EF=\frac{I_{SERS}}{I_{RS}}\times \frac{C_{RS}}{{С}_{SERS}},$$
where $I_{SERS}$ and $I_{RS}$—intensities of characteristic peaks of SERS and reference spectra, respectively; ${С}_{SERS}$ and $C_{RS}$—concentrations of analyte molecules, the signal of which was observed on the SERS and reference spectra, respectively.

## 3. Results and Discussion

In our previous work [48], we demonstrated the formation of an array of nanoparticles by two successive depositions on two sides of the pipette by vacuum–thermal evaporation of a 3 mg silver sample. The Ag nanoparticle array had an average diameter of about 16 nm (σ = 3.2 nm), the mean particle spacing was of the order of 8 nm with σ equal to 4.2 nm. The relative standard deviation σ/d and σ/L were 20% and 52%, respectively. To improve the homogeneity of the array, a technique using a pipette rotation during the evaporation of the sample was developed (Figure 1a). When the technique was applied to a rotating pipette with a 3 mg Ag sample and a rotation speed of 14 rps, the nanoparticle array had a mean diameter of about 8 nm (σ = 1.5 nm) and a particle spacing of the order of 6 nm (σ = 2 nm). The relative standard deviations σ/d and σ/L were 18% and 33%, respectively. As it can be seen, the relative standard deviation of σ/d and σ/L for the Ag nanoparticle array formed on the rotating pipette has a lower value. This indicates that the rotation provides greater uniformity in the spacing and homogeneity of the silver nanoparticle array, which is necessary to achieve high EF values as a key parameter of the SERS effect [49].

It is known that silver nanoparticles exhibit the phenomenon of plasmon resonance [24,50], which depends on the geometric parameters of the nanoparticle array [51,52] and has a direct relationship with the EF of SERS-active substrates [53]. Based on the equation [44], the effect of diameter and distance between Ag nanoparticles on the SERS electromagnetic enhancement factor was calculated for the case of two spherical nanoparticles (Figure 1b). As can be seen from Figure 1b, the EF value of the Raman signal reaches the highest values when the diameter of the nanoparticles increases and the distance between them decreases. For example, for a 5 nm particle spacing and a nanoparticle diameter of 25 nm, EF is of the 10^6^ order (Figure 1b). When applying vacuum–thermal evaporation to a rotating pipette, it is possible to control the diameter and spacing of the nanoparticles by varying the mass of the silver sample and the speed of rotation of the pipette, respectively.

At the first stage of the experiment, three silver-coated pipettes were produced using the silver samples of 3, 15 and 30 mg weight at the pipette rotation speed of 14 rps (Table 1). It should be noted that despite the increase in the particles’ spacing by 3 nm, the enhancement factor increased, because the particle size increased by 18 nm. The surface morphology of the pipette surface with deposited array of nanoparticles using 30 mg Ag sample at 14 rps is shown in Figure 1c. It should be noted that by using the same rotation speed and gradually increasing the Ag sample weight, there is an increase in the diameter and spacing of the particles. This trend must be taken into account to fabricate the SERS-active pipette with maximal EF. At the next step of the research, an experiment was carried out while evaporating a 30 mg Ag sample at different pipette rotation speeds to control the diameter and spacing of the particles. The geometric parameters of the silver nanoparticle arrays are given in Table 1. SEM images of each modified pipette and detailed information about size and particle spacings are demonstrated in the Appendix A.

Raman spectroscopy was used to determine the limit of detectable concentration using the modified pipettes. Figure 1f shows the SERS spectra of R6G adsorbed on the pipette tips. As it can be seen from Figure 1f, the following characteristic modes of R6G are present: 618 cm^−1^, 778 cm^−1^, 1190 cm^−1^, 1317 cm^−1^, 1365 cm^−1^, 1530 cm^−1^, 1578 cm^−1^, and 1653 cm^−1^ [54,55]. The concentration decrease leads to the quenching of some peaks. For example, at the concentration of 5 × 10^−8^ M (Figure 1f), just 4 peaks out of 8 initial peaks are present on the spectrum: 1190 cm^−1^, 1530 cm^−1^, 1578 cm^−1^, and 1653 cm^−1^. The limit of detection of R6G on the modified pipette is the concentration of 5 × 10^−8^ M.

Figure 3 shows the measured SERS spectra for R6G concentrations of 10^−3^ ÷ 5 × 10^−8^ M with modified pipettes, which were fabricated using 30 mg Ag samples and a rotation speed of 14 rps, and R6G 10^−1^ ÷ 10^−2^ M on an unmodified pipette for comparison. The peak observed at 1190 cm^−1^, co-corresponding to the mode C-H stretching of the xanthene ring of R6G, was chosen for the EF calculation [56]. Based on the I_SERS_ values, which are in the order of 307 a.u. for 5 × 10^−8^ M R6G on the Ag-modified pipette, and I_RS_ value of 794 a.u. for 10^−1^ M R6G applied on a no-particle pipette, the EF of the Raman signal was calculated. The EF value was about 7 × 10^5^. In addition, the peak observed at 1655 cm^−1^, co-corresponding to the C-C stretching mode of the xanthene ring of R6G, based on I_SERS_ values is of the order of 443 a.u. for 5 × 10^−8^ M R6G on the Ag-modified pipette [56]. I_RS_ = 1552 a.u. for 10^−1^ M R6G deposited on the pipette free of Ag nanoparticles. The EF value was about 5 × 10^5^. Thus, the calculated EF of the SERS-active pipette was about 7 × 10^5^. Such a large EF value indicates the high SERS activity of the modified pipette. Our EF is comparable to the designs of other researchers; a comparison is presented in the Appendix A. The significant EF provided by the Ag-modified pipette confirms its high suitability for the ultrasensitive detection and analysis of trace molecules. It can be seen from Figure 3 that the R6G SERS spectrum was successfully obtained for a concentration of 5 × 10^−8^ M. This fact confirms a significant increase in the efficiency of the modified pipette due to the application of our custom solution. This approach represents a new idea aimed at significantly extending the capabilities of conventional pipettes and SICM.

The modified pipettes fabricated with the 30 mg Ag sample and rotation speed of 14 rps were used as probes for SICM. Topographic images of the HEK-293 cell surface and maps of its elastomeric characteristics were thus obtained (Figure 2).

In this work, we compared the topography and Young’s modulus map obtained by non-modified and SERS-active pipettes. As a result, no prominent artifacts or noises were observed on the single-cell topography and Young’s modulus map obtained with SERS-active pipettes (Figure 2b) in comparison with non-modified ones (Figure 2a). Furthermore, statistical analysis (one-way ANOVA) did not reveal any significant differences between mean values of Young’s modulus obtained by SICM with non-modified and SERS-active pipettes (Figure 2c). The model shows that the differences between the non- and Ag-modified pipettes are negligible. Thus, altering the pipettes surface has no effect on the topography scanning and measurement of local mechanical parameters. The pipette modification for the SERS activity, as previously reported [45], was accompanied by full closure of the tip opening in the pipette, preventing scanning of the cell surface and, as a result, accurate control of cell distance and depth of indentation.

Local penetration of the modified pipettes into the nucleus and cytoplasm was accomplished via SICM, as illustrated schematically in Figure 2d. The fluorescent technique was used to determine the location of the nucleus and cytoplasm for local penetration of the SERS-active pipette, ensuring correct positioning (Figure 2e). The core has a higher contrast, as evidenced by the fluorescent pattern caused by the Hoechst dye. A darker region, on the other hand, corresponds to the cytoplasm. The SERS spectra from the produced modified pipettes were investigated in the following stage of the experiment, as illustrated schematically in Figure 2f.

The proposed method for the pipette modification without closure of its tip opening provides new prospects for single-cell analysis by SICM. As reported elsewhere [7], SICM can be used for correlative scanning of topography, Young’s modulus and confocal visualization. Thus, intracellular SERS measurements can be combined for correlative studying of cytoskeleton state as it is known that SERS spectroscopy is widely used for characterization of cytoskeleton bonds [57,58]. Similar methods were developed in combination with AFM [59]; however, cells were fixed due to high applied force and pressure. Moreover, precise positioning of the Ag-modified pipette used in SICM can potentially provide a collection of SERS spectra of different cell organelles, e.g., nucleus or mitochondria.

At the first step of the experiment of local identification of the nucleus and cytoplasm using Raman spectroscopy, SERS spectra of live cells and Hoechst dye placed on a SERS substrate [33] were obtained (Figure 3) for subsequent comparison with SERS spectra of the nucleus and cytoplasm from SERS pipettes. Green lines show the positions of peaks characteristic of the investigated HEK 293 cell: the 1004–1030 cm^−1^ peak belongs to phenylalanine [60,61]; 1174 and 1207 cm^−1^ peaks are characteristic for tyrazine, guanine, and phenylalanine [60,61,62]; 1275–1290 cm^−1^—amide III group [60,62]; the 1300 cm^−1^ peak is characteristic for the CH_2_ bending mode in proteins and lipids [62,63]; 1381 cm^−1^ CH_3_ bending; 1412 cm^−1^ is characteristic of asparagic and glutamic acids [60]; 1436 cm^−1^ is CH_2_ bending in lipids [62]; 1475–1485 cm^−1^ is amide group II [62]; 1524 cm^−1^ is carotenoids not characteristic of normal tissues [62]; 1550 cm^−1^ is tryptophan [62]; 1578 cm^−1^ is guanine and adenine [60,62]; 1600–1650 cm^−1^ is amide group I [62] (Appendix A). As can be seen from Figure 3, the SERS spectra of the cytoplasm and nucleus contain groups of peaks of phenylalanine, amide I, amide III, guanine, adenine, proteins, and lipids, which correspond to the peaks in the SERS spectrum of living cell HEK 293. The procedure of SICM scanning and obtaining SERS spectra took about 20 min.

Orange lines in Figure 3 show the positions of peaks characteristic of Hoechst-980, 1358, 1456, 1610 cm^−1^ [64], which are not observed in the SERS spectrum of living cells. It is worth noting that these peaks are present in the SERS spectrum of the nucleus obtained from the SERS pipette. This is explained by the penetrating ability and high binding to DNA of the dye, which mainly accumulates in the nucleus [65]. It also draws attention to the fact that Hoechst peaks are not observed in the SERS spectrum of the cytoplasm. The local penetration of the SERS pipette into the nucleus is confirmed by the presence of an intense peak in the range 1322–1337 cm^−1^ (Figure 3) (blue spectrum), which is characteristic mainly for vibrations of DNA nucleic acid fragments [62], which are part of chromosomes. It should be noted that, under normal conditions, DNA is absent in cytoplasm, which is confirmed by the absence of the peak in the region 1322-1337 cm^−1^ for the SERS spectrum of cytoplasm (red spectrum). At the same time, the peak at 1337 cm^−1^, characteristic for adenine and guanine, which are components of nucleic acids that make up the nucleus, is present in the SERS spectrum of living cells. An additional fact confirming the local penetration of the pipette into the nucleus is the guanine peak (1174 cm^−1^), which has a higher intensity in the SERS spectrum of the nucleus than of the cytoplasm [62].

Based on the obtained results of SERS studies, we can conclude that a credible fact of local penetration of the modified pipette with SICM into the cell nucleus is the presence of a group of peaks of nucleic acids (guanine, and adenine) and dye in the SERS spectra of the nucleus specific for this part of the cell. The corresponding groups of peaks of cytoplasm are absent in the SERS spectrum. Thus, these statements prove the possibility of local identification of different parts of cells using pipettes, the surface of which is modified by an SERS-active layer based on Ag nanoparticles.

## 4. Conclusions

In this study, the possibility of forming a homogeneous SERS-active array of Ag nanoparticles on the surface of a pipette by using vacuum–thermal evaporation on the rotating surface of the sample is demonstrated. Based on the results of the mathematical model, an array of silver nanoparticles with optimal geometrical parameters and theoretically possible Raman signal enhancement factor was selected, which was EF = 5 × 10^4^ for Ag nanoparticles with a diameter of 36 nm and particle spacing of 12 nm. SERS studies of the obtained sample are presented, which demonstrated a detection limit of R6G of 5 × 10^−8^ M and a Raman signal enhancement factor of 7 × 10^5^, which is slightly higher than the theoretically predicted EF. Applications of SERS-active pipettes in a scanning ion-conducting microscope to estimate Young’s Modulus have shown that the mean values are 0.76 ± 0.08 kPa and 0.89 ± 0.09 kPa for non-modified and SERS pipettes, respectively. SERS-active pipettes and Raman spectroscopy have the ability to locally identify the location of the nucleus and cytoplasm of living cells and to detect their molecular composition of cytoplasm and nucleus was demonstrated, confirming the high sensitivity of the modified pipette. SERS studies showed the presence of a group of peaks of nucleic acids (guanine, and adenine) and Hoechst dye in the SERS spectrum of the nucleus, and their corresponding absence in the SERS spectrum of the cytoplasm. The presented results will expand the research potential of SICM and will help obtain new data for the study of living cells.

An effect of the biological environment is of a great research interest but this must be a separate comprehensive study since the cell organelles are characterized by rather different molecular composition, which can contain molecules with S-S, S-H bonds broken in the presence of silver. As a result, such molecules are chemisorbed on the silver surface providing an enhanced signal from specific cell components. Therefore, an outlook of the present study is engineering the SERS measurements protocol to obtain reproducible and reliable results including the interaction of the silvered pipette with the cell environment.

## Figures and Tables

**Figure 1 cells-12-02521-f001:**
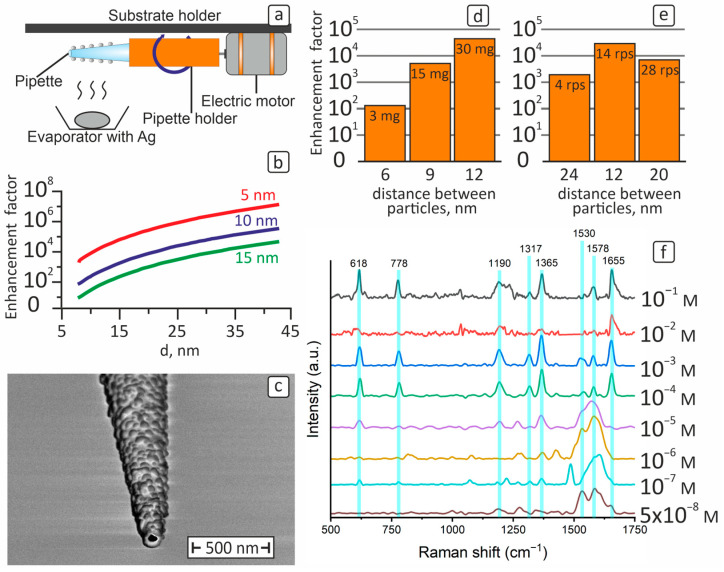
(**а**) Schematic representation of the process of vacuum–thermal evaporation of Ag on a rotating pipette; (**b**) calculated dependence of the SERS electromagnetic component of EF as a function of the diameter of nanoparticles and the distances of 5 nm (red), 10 nm (blue), 15 nm (green) between them; (**c**) surface morphology of the pipette modified with silver nanoparticles during the evaporation of a 30 mg Ag sample at a speed of 14 rps; (**d**) calculation of predicted EF for modified pipettes at 3, 15, 30 mg evaporation at a speed of 14 rps; (**e**) calculation of the predicted EF for the modified pipettes for evaporation of a 30 mg sample at speeds of 4, 14, 28 rps; (**f**) SERS spectra of R6G for 10^−3^ ÷ 5 × 10^−8^ M obtained from the tips of modified pipettes and spectra of R6G-from the clean pipettes for 10^−1^ ÷ 10^−2^ M (black and red spectra).

**Figure 2 cells-12-02521-f002:**
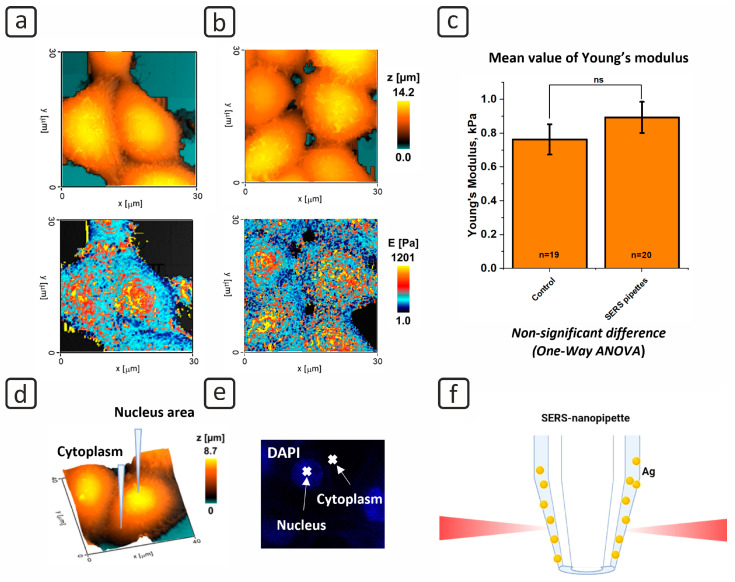
(**a**) Topography (above) and Young’s modulus map (below) obtained by non-modified pipettes; (**b**) topography (above) and Young’s modulus map (below) obtained by the SERS-active pipettes; (**c**) mean value of Young’s modulus (n—the number of measured cells; ns—non-significant difference); (**d**) scheme of local cell penetration and sampling for SERS; (**e**) fluorescence of cell nucleus (Hoechst dye); (**f**) obtaining SERS spectrum from the sampled cell.

**Figure 3 cells-12-02521-f003:**
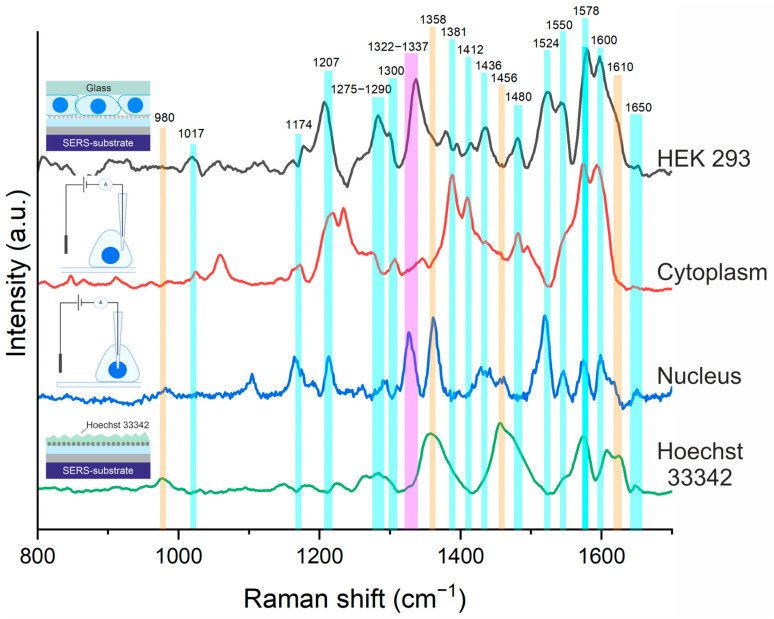
SERS spectra of HEK 293 (black), Hoechst 33342 dye (green) obtained using SERS substrate [33], and SERS spectra of cytoplasm (red) and cell nucleus (blue) obtained using a pipette modified with Ag nanoparticles that locally penetrated the corresponding parts of HEK 293.

**Table 1 cells-12-02521-t001:** Geometrical parameters of the silver arrays when depositing 3, 15, 30 mg samples and pipette rotation speeds of 4, 14, 28 rps on pipettes.

Weight of Ag, mg	Rotation Speed, rps	Diameter of Particles, nm	D, nm^2^	σ, nm	Particles Spacing, nm	D, nm^2^	σ, nm
3	14	8	2.4	1.5	6	4.2	2
15		18	33.5	5.8	9	4.6	2.2
30		36	47.8	6.9	12	15	3.9
30	4	42	123.3	11.1	24	57.7	7.6
	14	36	47.8	6.9	12	15	3.9
	28	45	86.3	9.3	20	78.9	8.9

## Data Availability

Not applicable.

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
