# Peer review of "Single-Cell Analysis with Silver-Coated Pipette by Combined SERS and SICM"

_cells, 2023, doi:10.3390/cells12212521_

Round 1

Reviewer 1 Report

n this MS authors have presented significant results on an application of surface-enhanced Raman spectroscopy (SERS) and scanning ion-conducting microscopy (SICM) for single cell analysis.

Although the work showed benefits of the silver-coated pipette for both the SICM and the molecular composition analysis of cytoplasm and nucleus using SERS method, the MS needs to be revised. See comments below.

Some comments:

1.            Authors need to provide the statistical analysis and related details in the abstract/results/conclusions sections.

2.            Authors should clearly define the advantages of their approach over preexisting tools such as fluorescence-based imaging platforms.

3.            Authors could refer to some other studies involving SERS for single cell analysis in plants/bacteria, etc. See: PMID: 28301688 and PMID: 22938600

4.            Provide proper citations for tip fabrication techniques.

5.            For figure 3…how spectra were assigned could be explained in the MS.

6. How long does it take to acquire should be explained in abstract and results?

7. Could this approach be utilized in clinical settings?

Moderate editing is necessary for clarity.

Author Response

Dear reviewer, Thank you for your helpful suggestions and comments. We have been improving the manuscript with your comments in mind. Thank you for the fruitful discussion! Below are the responses to your comments.  In the document with the answers I also enclose the supplementary material.

1.Authors need to provide the statistical analysis and related details in the abstract/results/conclusions sections.

Thank you for your suggestion. We have been improving the manuscript and have brought in additional information, including:

-added statistical details to the abstract (Line 13);

-expanded Table 1 (Line 281) with statistical data on the processing of all modified pipette samples. And also added supplementary material, which shows additional data on SEM images and diagrams of size distribution of diameter and particle spacing.

- The supplementary material shows extended data on obtained SERS-spectra for live cells.

2.Authors should clearly define the advantages of their approach over preexisting tools such as fluorescence-based imaging platforms.

Thank you for your helpful comment. We have improved the manuscript and added additional information to the "introduction" (Line 39):

The combination of a fluorescence-based imaging platform and SERS is frequently employed for researching cell functional characteristics and their interactions with nanomaterials (10.1039/C1NR11243K; 10.1039/C7AY02039B; 10.1016/j.apsb.2022.08.024). However, resolution of such methods are limited by half of the excitation wavelength. The SERS application of SICM based on silver-coated nanopipette provides possibility for scanning topography with nanoscale resolution prior to the SERS sampling procedure. The feedback control of SICM based on ion current registration (10.1021/acs.chemrev.0c00962) allows to perform SERS sampling on different compartments of living cells with highly controlled depth. Also, in our previous work (10.1039/D0NR08349F) we demonstrated SICM application for Young’s modulus estimation of living cells based on intrinsic force between nanopipette tip and cell surface. In this work we revealed that the same characterization of local mechanical properties can be performed by using silver-coated nanopipette for SERS.

3. Authors could refer to some other studies involving SERS for single cell analysis in plants/bacteria, etc. See: PMID: 28301688 and PMID: 22938600

Thank you for your suggestion. We have added the appropriate reference to the manuscript - [10.1111/tpj.13537,10.1016/j.aca.2012.07.037] (Line 76):

In this case, a pipette plays a role of SERS-active tool [13, 14, 15]. SERS-technique have also been actively used to study single cells of plants and bacteria [10.1111/tpj.13537, 10.1016/j.aca.2012.07.037].

4. Provide proper citations for tip fabrication techniques.

Thank you for your suggestion. We have added the appropriate reference to the manuscript (Line 123):

Kolmogorov, V.S. et al., 2021. Mapping mechanical properties of living cells at nanoscale using intrinsic nanopipette–sample force interactions. Nanoscale 13, 6558–6568. https://doi.org/10.1039/D0NR08349F

5. For figure 3…how spectra were assigned could be explained in the MS.

Thanks for the helpful comment. We have added relevant tables to the supplementary material in this paper (Table 2,3)

6. How long does it take to acquire should be explained in abstract and results?

Thank you for this useful suggestion. SERS sampling doesn’t take a long time. Scanning procedure of the cell takes 2-3 minutes by SICM. Then, SERS sampling by silver-coated nanopipette takes 10 minutes. The SERS investigation took no more than 5 minutes to complete. From the procedure of scanning to taking the SERS spectrum took about 20 minutes. We have improved the manuscript and added information (line 357).

7. Could this approach be utilized in clinical settings?

Thank you for your suggestion. We have improved the manuscript and added information to the introduction section (Line 111):

In this study we have developed a technique for the formation of an array of Ag nanoparticles by thermal vacuum evaporation onto a rotating pipette for SICM studies of living cells and for the accurate differentiation of the pipette location in cell and ultrasensitive detection of anomalies in molecular composition of membrane, cytoplasm and nucleus depending on a target cell line or external effects like interaction with drugs and temperature variation, to name a few. In previous works [10.1016/j.ejmech.2021.113936; 10.1021/acs.jmedchem.1c01157] we showed that SICM can be successfully used for characterization of novel anticancer drugs such as cytostatics.Because the mechanism of cytostatics is a change in the cytoskeleton state, the efficiency of a novel medicine can be estimated by measuring the cell Young's modulus. Thus, SERS-specific analytes can be characterized simultaneously in a single cell using a combination of silver-coated nanopipettes.

Reviewer 2 Report

Silver-coated peptide nanoparticles were synthesized and used for cell imaging by using SERS technique. The subject is valuable, interesting and publishable. However, there are some points which should be addressed, clarified and/or discussed by the authors before the final publication. Therefore, I suggest revision of the manuscript according to the following comments:

1.       In Table 1, the size of silver nanoparticles is mentioned. These sizes are in the range of 8-45 nm. I can understand the estimation of the maximum size by using a typical SEM image similar to what is presented in Figure 1c. But what about the minimum sizes such as 8 and 6 nm? This subject should be clarified in the revised version. In addition, the values need to be presented based on their statistical and/or technical dispersions.   

2.       It has been mentioned that “Raman scattering (SERS) spectroscopy, which unique sensitivity is known to provide detection and identification of single molecules”. This statement can be further supported by a recent work such as [doi.org/10.1007/s11468-023-02042-1].

3.       No evidence relating to the plasmonic property of the silver nanoparticles can be found in the manuscript. Optical absorption/reflection measurement can be used in this regard. In addition, the position of the surface plasmon resonance peak can be used for evaluation of the size of the metallic nanoparticles. See, for example, [Nanotechnology 17 (2006) 763]. This subject should be addressed and discussed in the revised version.  

4.       It has been mentioned that “An immersion deposition, which is a derivative of the chemical approach, has been exploited for careful adjusting the geometrical parameters of metal nanoparticles [24, 25]”. However, ref. 24 is related to Cu deposition rather than Ag, as the main subject of the manuscript. Hence it can be replaced by a more suitable one such as [Spectrochimica Acta Part A: Molecular and Biomolecular Spectroscopy Volume 298, 5 October 2023, 122762].

5.       Figure 1c shows a cone like feature which can act as a rod like antenna with especial surface plasmon resonance property. The properties of nanosilver-based antennas suitable for biological imaging was previously reported in [Optical Engineering 58 (2019) 065102]. This subject should be addressed and discussed in the revised version.

6.       The chemical state of silver nanoparticles can change in exposure to air and/or biological media. This can affect the sensitivity of the and the accuracy of the images. Could the authors comment on the stability in the performance of the process after a long time or many cycles?

7.       Figure 1d shows that by increasing the distance between the particles the EF increased. But, Figure 1b presents that increasing the distance between particles resulted in decreasing the EF. Why? This issue should be clarified and discussed in the revised version.    

8.       Could the authors compare the EF obtained in this work with the EFs obtained in the previous related works? A table can be designed in this regard.

Some minor revisions can be considered. 

Author Response

Dear reviewer, Thank you for your helpful suggestions and comments. We have been improving the manuscript with your comments in mind. Thank you for the fruitful discussion! Below are the responses to your comments. 

In the document with the answers I also enclose the supplementary material.

Yours sincerely,

Dr. Sergey Dubkov, Aleksei Overchenko, on behalf of the authors.

1.In Table 1, the size of silver nanoparticles is mentioned. These sizes are in the range of 8-45 nm. I can understand the estimation of the maximum size by using a typical SEM image similar to what is presented in Figure 1c. But what about the minimum sizes such as 8 and 6 nm? This subject should be clarified in the revised version. In addition, the values need to be presented based on their statistical and/or technical dispersions.

Thank you for your very helpful comment. We agree with it and have improved the manuscript. SEM was used to estimate the geometric parameters and subsequent processing of the acquired images in ImageJ program. We have improved the manuscript and added additional information in the methods section (Line 155); expanded Table 1 and added information on the statistical analysis of other modified pipette samples (Line 280). Supplementary material was added to this work, in which SEM images of the obtained samples are shown in Figure 1; Figures 2 and 3 show the distribution diagrams of nanoparticle diameters and spacing. As can be seen in the enlarged image of Figure 1a(supplementary material) there is an array of silver nanoparticles with the corresponding sizes shown by us in the manuscript.

2. It has been mentioned that “Raman scattering (SERS) spectroscopy, which unique sensitivity is known to provide detection and identification of single molecules”. This statement can be further supported by a recent work such as [doi.org/10.1007/s11468-023-02042-1].

Thank you for your useful comment. We have improved the manuscript and added the reference you provided.

3. No evidence relating to the plasmonic property of the silver nanoparticles can be found in the manuscript. Optical absorption/reflection measurement can be used in this regard. In addition, the position of the surface plasmon resonance peak can be used for evaluation of the size of the metallic nanoparticles. See, for example, [Nanotechnology 17 (2006) 763]. This subject should be addressed and discussed in the revised version.  

Thank you for your helpful comment. We agree that an investigation of the plasmon resonance position would strengthen this work. However, it is worth noting that the formation of silver nanoparticles was performed on pipette tips that are cone-shaped and nanometer-sized. Conducting a plasmon resonance position study on such samples is a very time consuming task and would require the development of a new absorption/reflection study technique to obtain reliable information. On the other hand, the plasmonic properties of silver and other nanostructures are now well studied and confirmed[10.1155/2009/475941; 10.1007/s00396-018-4308-9; 10.1038/s41598-018-36491-0; 10.1002/jrs.4093], as well as in the work cited by the distinguished reviewer [10.1088/0957-4484/17/3/025]. We have improved the manuscript and added information (Line 249):

It is known that silver nanoparticles exhibit the phenomenon of plasmon resonance [16, 10.1088/0957-4484/17/3/025], which depends on the geometric parameters of the nanoparticle array [42, 43] and has a direct relationship with the EF of SERS-active substrates [10.1039/C7NR08959G].

We agree with the second part of your comment that nanoparticle size determination using the plasmon resonance position is possible. In this work, we used SEM to determine the diameters of silver nanoparticles of all experimental samples, which allows us to obtain not only information about the diameter of nanoparticles, which is possible to obtain by analyzing the plasmon resonance position, but also to collect the necessary statistical information, which was included in the supplementary material (Figure 1,2,3).

4. It has been mentioned that “An immersion deposition, which is a derivative of the chemical approach, has been exploited for careful adjusting the geometrical parameters of metal nanoparticles [24, 25]”. However, ref. 24 is related to Cu deposition rather than Ag, as the main subject of the manuscript. Hence it can be replaced by a more suitable one such as [Spectrochimica Acta Part A: Molecular and Biomolecular Spectroscopy Volume 298, 5 October 2023, 122762].

Thank you for your useful comment. We have changed [24] to the reference you provided.

5. Figure 1c shows a cone like feature which can act as a rod like antenna with especial surface plasmon resonance property. The properties of nanosilver-based antennas suitable for biological imaging was previously reported in [Optical Engineering 58 (2019) 065102]. This subject should be addressed and discussed in the revised version.

Thank you for your suggestion. We have added the appropriate reference to the manuscript - 10.1117/1.OE.58.9.097102(Line 109):

Due to the nanoscale tip of the pipette (10-100 nm), it has minimal effect on the cell under study when it is inserted into the cell [28]. A similar cone-shaped structure which act as a rod like antenna with especial surface plasmon resonance property is mentioned in work [10.1117/1.OE.58.9.097102].

6. The chemical state of silver nanoparticles can change in exposure to air and/or biological media. This can affect the sensitivity of the and the accuracy of the images. Could the authors comment on the stability in the performance of the process after a long time or many cycles?

Thank you for your helpful comment. Yes we agree with this suggestion. Silver nanoparticles are known to be susceptible to sulfidization, which entails deterioration of the SERS effect. In this case, the modified pipettes were stored in a vacuum box and were not exposed to external atmosphere. We have improved the manuscript and expanded the description of the "materials and methods" section (line 111):

 Before and after the nanoparticle deposition process, the pipettes were stored in vacuum at room temperature to prevent contamination of the pipette, degradation of the particles array [31]. The vacuum box with modified nanopipettes was opened immediately before each measurement with cells. In this way we avoided surface composition changes caused by air and provided the same storage conditions prior the measurements.

Regarding the second part of your comment, it is important to note that it is not possible to study pipette stability within the framework of this study, as the process will take a long period of time. In our opinion, the pipette stability study requires an additional separate study that will be conducted by our group in the future. We have added a corresponding improvement to the "conclusions" section (line 409)

An effect of the biological environment is of a great research interest but this must be a separate comprehensive study since the cell organels are characterized by rather different molecular composition, which can contain molecules with S-S, S-H bonds broked in silver presence. As a result, such molecules are chemisorbed on the silver surface providing an enhanced signal from specific cell components. Therefore an outlook of the present study is engineering the SERS-measurements protocol to obtain reproducible and reliable results including interaction of the silvered pipette with cell environment.

7. Figure 1d shows that by increasing the distance between the particles the EF increased. But, Figure 1b presents that increasing the distance between particles resulted in decreasing the EF. Why? This issue should be clarified and discussed in the revised version.

Thank you for your comment. You are right about figure 1b. However, it should be noted that the enhancement factor is related to both the distance between the particles and the particle size [https://doi.org/10.1155/2015/124582, https://doi.org/10.1117/12.2511299]. Yes, it can be seen from Figure 1d that the enhancement factor increases despite the increase in the distance between the particles. But there is a significant increase in particle size with increasing Ag sample, notice Table 1. The distance between the particles gradually increased by 3 nm, but at the same time the size of the nanoparticles increased by several times. Thus, due to the strong increase in size, the small increase in the distance between particles is not so critical for the case of the modified pipette, therefore the enhancement factor increased. We have added a relevant discussion and improved the manuscript (Line 268).

At the first stage of the experiment, three silver-coated pipettes were produced using the silver samples of 3, 15 and 30 mg weight at the pipette rotation speed of 14 rps (Table 1). It should be noted that despite the increase in the particles spacing by 3 nm the enhancement factor increased, because the particle size increased by 18 nm.

8. Could the authors compare the EF obtained in this work with the EFs obtained in the previous related works? A table can be designed in this regard.

Thanks for the helpful comment. We agree and have added Table 1 to the supplementary materials for this manuscript. Table 1 shows a comparison of the enhancement factors of the SERS-structure obtained in this work and the closest SERS-structures presented in the works of other researchers. Based on the obtained values, it can be seen that the EF of the structure from this work is comparable to the close analogs.

Round 2

Reviewer 2 Report

The authors well revised the manuscript based on the comments. However, there is a minor point which should be considered before the final publication, as mentioned below:

Concerning Comment #5, the response is "Thank you for your suggestion. We have added the appropriate reference to the manuscript". But, it seems that ref. 36 is not consistent with the reference introduced in the comment. For further help to the authors, I add its doi as follows:  doi.org/10.1117/1.OE.58.6.065102

After considering this point the manuscript is publishable. 

1.       doi.org/10.1117/1.OE.58.6.065102

can be improved during the final revision. 

Author Response

Dear Reviewer,
Thank you for providing this information. We have corrected the reference [36].